# Combining spectroscopic and isotopic techniques gives a dynamic view of phosphorus cycling in soil

Julian Helfenstein [1], Federica Tamburini[1], Christian von Sperber[2,3], Michael S. Massey[4], Chiara Pistocchi[5], Oliver A. Chadwick[6], Peter M. Vitousek[7], Ruben Kretzschmar[8] & Emmanuel Frossard[1]

Current understanding of phosphorus (P) cycling in soils can be enhanced by integrating previously discrete findings concerning P speciation, exchange kinetics, and the underlying biological and geochemical processes. Here, we combine sequential extraction with P *K*-edge X-ray absorption spectroscopy and isotopic methods ($^{33}$P and $^{18}$O in phosphate) to characterize P cycling on a climatic gradient in Hawaii. We link P pools to P species and estimate the turnover times for commonly considered P pools. Dissolved P turned over in seconds, resin-extractable P in minutes, NaOH-extractable inorganic P in weeks to months, and HCl-extractable P in years to millennia. Furthermore, we show that in arid-zone soils, some primary mineral P remains even after 150 ky of soil development, whereas in humid-zone soils of the same age, all P in all pools has been biologically cycled. The integrative information we provide makes possible a more dynamic, process-oriented conceptual model of P cycling in soils.

---

[1] Institute of Agricultural Sciences, ETH Zurich, 8315 Lindau, Switzerland. [2] Institute of Crop Science and Resource Conservation, University of Bonn, 53115 Bonn, Germany. [3] McGill University, Montreal QC H3A 0B9, Canada. [4] Department of Earth and Environmental Sciences, California State University East Bay, Hayward, CA 94542, USA. [5] Eco&Sols, Montpellier SupAgro, University of Montpellier, CIRAD, INRA, IRD, 34060 Montpellier, France. [6] Department of Geography, University of California, Santa Barbara, CA 93106, USA. [7] Department of Biology, Stanford University, Stanford, CA 94305, USA. [8] Institute of Biogeochemistry and Pollutant Dynamics, ETH Zurich, 8092 Zürich, Switzerland. Correspondence and requests for materials should be addressed to J.H. (email: julian.helfenstein@usys.ethz.ch)

Phosphorus (P) cycling in soils is a basis for many ecosystem services, including food production, water quality regulation, and carbon sequestration[1,2]. Hence, there is growing demand to incorporate P cycling into Earth-system models to improve their ability to predict ecosystem response to global changes. To understand P cycling in soils and pave the way to incorporating P into Earth-system models, the large number of often-poorly defined inorganic and organic P species in soils have to be compartmentalized into a manageable and meaningful set of pools, the kinetics of exchange processes between these pools and the soil solution have to be quantified, and biological processes must be integrated with geochemical processes[1]. A diverse set of spectroscopic and isotopic techniques are available to address these challenges[3], yet few studies have integrated the strengths of different approaches to address all three.

The most common way to compartmentalize soil P is through sequential extractions, which yields operationally defined pools that are used not only to assess soil fertility and soil development[4–6], but also as a basis for modeling soil P dynamics[7–9]. Commonly considered pools examined in this study are resin-extractable P, hexanol–resin fumigation extraction as microbial P, 0.25 M NaOH and 0.05 M EDTA- extractable inorganic (NaOH-Pi) and organic P (NaOH-Po), 1 M HCl-extractable P, and the remaining P as residual P (see Methods for details). While it is often assumed that pools from sequential extractions contain distinct forms of P, the composition of extracted pools is hypothetical[10,11]. Furthermore, it is often assumed that certain pools are bioavailable, while others are not, even though P is exchanged among all pools. To translate information on P pools into information about P availability, the kinetics of such exchanges must be determined.

Understanding the dynamics of P cycling presents us with a dichotomy: at the scale of the soil profile, P cycling operates at timescales of centuries–millennia, while P in the soil solution turns over within seconds[12]. Yearly inputs to soil through weathering and dust deposition are on the order of $10^{-4}$ to $10^{-1}$ mg P kg$^{-1}$ yr$^{-1}$ (refs.[13,14]), while total soil P stocks are generally on the order of $10^1$–$10^3$ mg P kg$^{-1}$ (refs.[6,15]). However, at the solid-solution interface, $10^{-1}$–$10^2$ mg P kg$^{-1}$ may be exchanged in 1 min[12,16].

Microbes and plants drive P cycling by mineralizing organic P, taking up P, synthesizing new organic P, and affecting the solubility of P minerals through exudates[17,18]. Stable oxygen isotope ratios in phosphate ($\delta^{18}O_P$) can provide information on biological processes, because under soil conditions, only enzymes can break the bond between P and O in phosphate and alter the $\delta^{18}O_P$[19]. Previous studies with this technique have revealed the role of microbes in P cycling in different sequentially extracted pools[20,21], and have also shown that phosphate in plants is often more enriched in $^{18}O$ than phosphate in soil[22]. Isotopic fractionation factors for the hydrolysis of organic P by key soil enzymes have been determined[23–25]. Thus, $\delta^{18}O_P$ can be used to differentiate between phosphate derived from geological parent material and P that has cycled through microbes or plants, and has the potential to provide information on dominant enzymatic processes.

Here, we applied sequential extraction together with spectroscopic and isotopic analyses to integrate information on P pools, turnover times, and biological processes. We used sequential chemical extraction to measure the size of operationally defined soil P pools. We compared the sequential extraction to classes of soil P, as determined by P K-edge X-ray absorption near-edge fine structure (XANES) spectroscopy. Furthermore, we measured P exchange in the soil solid-solution interface using $^{33}P$, and compartmentalized soil P into P that is exchangeable within discrete time steps. These P pools defined by kinetics were then correlated to P pools, as determined by sequential extraction, to approximate turnover times. Finally, we measured $\delta^{18}O_P$ in unweathered parent material and plants, as well as in resin, microbial, NaOH-Pi, NaOH-Po, and HCl soil pools, allowing us to trace imprints of biological processes through the P cycle.

We applied our approach to the well-studied Kohala climatic gradient on the Island of Hawaii, where soil developed on ca. 150,000-year-old lava flow[26–29] (Supplementary Table 1). On this gradient, soil age and parent material are constant, while rainfall ranges from 280 to 3100 mm over only 12 km[30]. On arid sites, plants only partially cover the soil and nutrient-rich top soil may be removed by wind erosion, while on humid sites, mean annual precipitation is well above potential evapotranspiration, leading to intensive leaching[31]. An earlier study documented an increase in soil organic matter and a transition from primary to secondary and amorphous minerals with increasing rainfall[27] (Supplementary Fig. 1). Based on non-linear changes in soil mineralogy and nutrient concentrations, the gradient has been divided into three soil process domains[26], here called arid, subhumid, and humid.

## Results

**Distribution of P pools**. Soil total P was the highest on subhumid sites, and the lowest at the wettest site (Supplementary Table 2). The relative size of soil P pools changed predictably with increasing precipitation, with microbial P increasing while resin-P decreased. Also, the relative concentration of HCl-P declined, while NaOH-extractable Pi and Po pools increased in wetter sites (Fig. 1). The HCl-P pool is commonly considered to be composed of apatite[32]. With bulk XANES, apatite was only observed on the arid end of the climatic gradient. In subhumid and humid sites, the spectra could be fit using only the P on hematite and P Al-oxide-humic complex reference spectra, suggesting that P was dominantly associated to Al or Fe, or in organic form (Supplementary Fig. 2, Supplementary Table 3). Hence, the XANES results corroborate that the HCl pool contains apatite on dry sites, but at subhumid and humid sites, apatite is not present in a large enough quantity to show up on the bulk XANES. There, the HCl pool must contain P from other P forms.

**P isotope exchange kinetics**. Recovery of $^{33}P$ in the soil solution 1 min after addition dropped from $49 \pm 2\%$ at the driest site to $7 \pm 0.3\%$ at the wettest site (Fig. 2a). The dilution of radioisotopes was modeled with

$$\frac{r_{(t)}}{R} = m\left(t + m^{\frac{1}{n}}\right)^{-n} + \frac{r_{(\infty)}}{R} \qquad (1)$$

by fitting $m$ and $n$, where $r_{(t)}$ is the radioactivity measured at time $t$, and $R$ is the total amount of radioactivity added to the solution, and $\frac{r_{(\infty)}}{R}$ is set equal to the proportion of water extractable to total inorganic P[16].

Water-extractable P ($P_w$) followed the pattern of total P, in that it generally decreased with increasing precipitation but peaked at subhumid site 3, and contained 0.06–0.9% of total P at all sites. The mean turnover of P in the soil solution was between 1.9 and 59 min$^{-1}$, tending to increase with increasing precipitation (Supplementary Table 4). Based on the dilution of radioisotopes $r_{(t)}/R$ and soil solution P concentration, physicochemical P exchange as a function of time ($E_{(t)}$) can be calculated using Eq. 2[33].

$$E_{(t)} = P_w \times \frac{R}{r_{(t)}} \qquad (2)$$

Site 3 had the highest amount of exchangeable P ($E_{(t)}$) at all times, in line with high total P at this site (Fig. 2b). $E_{(t)}$ increased

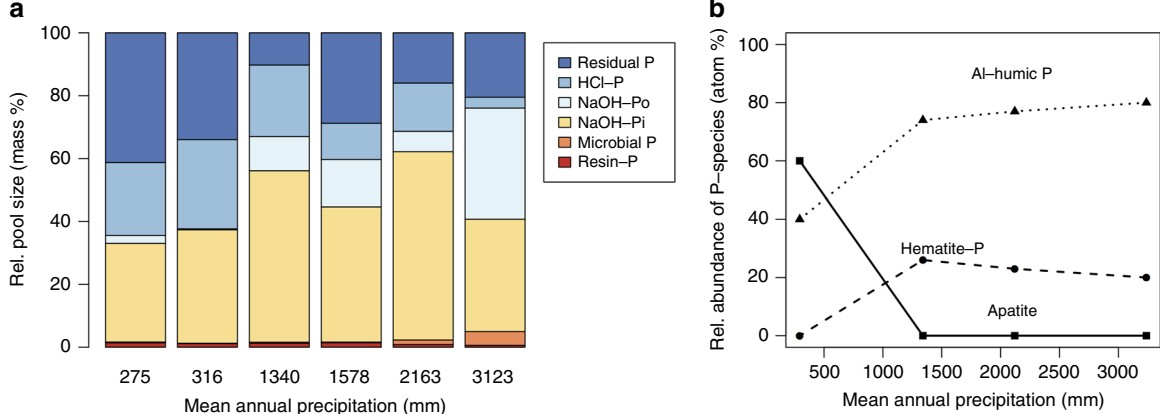

**Fig. 1** Phosphorus pool distribution along the climatic gradient. Relative concentration of P pools, as determined by sequential extraction (**a**) and of P classes by bulk P K-edge X-ray absorption near-edge structure (XANES) (**b**). Measured concentrations are in the supplemental material (Supplementary Table 2). XANES was performed at four sites along the gradient (**b**). XANES relative abundances are accurate to ±10% with a detection limit of around 10%

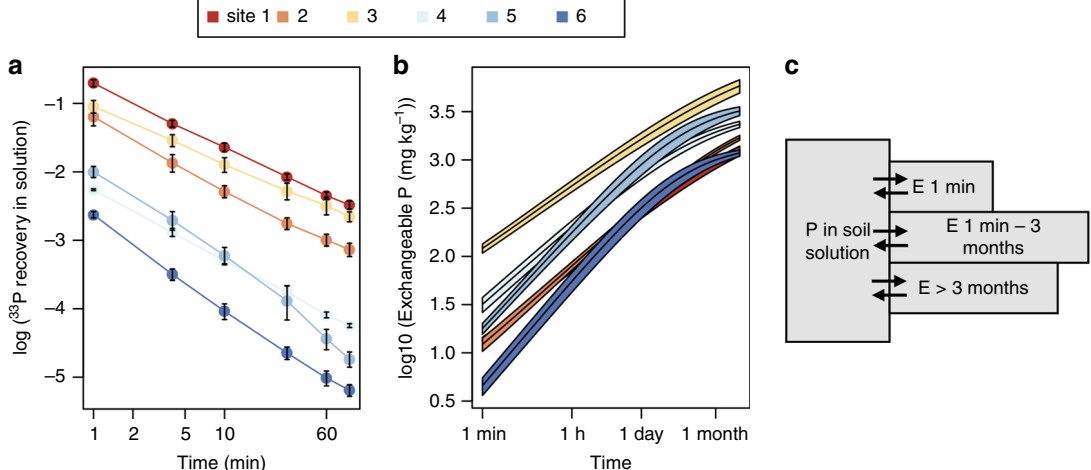

**Fig. 2** Radioisotope dilution and P exchangeability. The fraction of $^{33}$P recovered in the soil solution in minutes after addition $r(t)/R$ (**a**). Isotopically exchangeable P ($E_{(t)}$) for time spans up to 3 months was calculated as $E_{(t)} = P_w \times \frac{R}{r_{(t)}}$, where $P_w$ is the concentration of P in the soil solution (**b**). Error ranges in both plots represent one standard deviation of four values. Conceptual representation of how $E_{(t)}$ is divided into time-dependent isotopically exchangeable P pools (**c**). Sites 1–6 have mean annual precipitations of 275, 316, 1340, 1578, 2163, and 3123 mm yr$^{-1}$, respectively

more quickly in soil from humid than from arid sites. On humid sites, $E_{(t)}$ increased quickly at short times but flattened off at about 1 month, whereas on arid sites, $E_{(t)}$ increased more gradually throughout the time frame.

Using $E_{(t)}$, inorganic soil P was binned into pools depending on exchange time (Fig. 2c). Time-dependent $E_{(t)}$ correlated with pools from the sequential extraction. Resin-P correlated with $E_{(1 \, min)}$, NaOH-Pi with $E_{(3 \, months)} - E_{(1 \, min)}$, and HCl-P with $E_{(>3 \, months)}$ (Fig. 3).

**Oxygen-stable isotopes in phosphate**. The theoretical equilibrium window for $\delta^{18}O_P$ in soil resulting from cycling by pyrophosphatases was 20.5–29‰ for arid, 16.4–22.6‰ for subhumid, and 18.4–22.4‰ for humid sites. These theoretical equilibria were calculated considering temperature ($T$ [K]) and the $\delta^{18}O$ value measured in soil water (Eq. 2)[34]:

$$\delta^{18}O_P = e^{\left(\frac{14.43}{T} - 0.0265\right)} \times \left(\delta^{18}O_{H_2O} + 1000\right) - 1000 \quad (3)$$

Plant water was more enriched in $^{18}O$, and accordingly, the theoretical equilibrium for plant phosphate was also higher

(Supplementary Table 5). Using the fractionation factors of acid and alkaline phosphatase and phytase, measured $\delta^{18}O_P$ of organic P, and $\delta^{18}O$ of water, we calculated that phosphate released by the breakdown of soil organic P by one of these enzymes would have $\delta^{18}O_P$ in the range of 6.3–18.3‰ (Fig. 4a) (Supplementary Table 6). These theoretical calculations were used to assist in the interpretation of measured values of $\delta^{18}O_P$ in soil P pools.

Measured oxygen-stable isotope ratios in phosphate were the lowest in parent material and the highest in plant samples, with soil P pools ranging in between. Parent material had a $\delta^{18}O_P$ of 10‰. With a value of 23–35‰, TCA-extractable plant P represented a source of heavy phosphates (Supplementary Table 5). The $\delta^{18}O_P$ of almost all soil P pools was within the soil equilibrium range or slightly above (Fig. 4b, c). $\delta^{18}O_P$ of NaOH-Po was consistently around 20‰ and generally in the range of $\delta^{18}O_P$ of NaOH-extractable plant P. Two soil pools had substantially below-equilibrium $\delta^{18}O_P$ values: HCl-P in arid sites and microbial P in humid sites. HCl-P $\delta^{18}O_P$ in the B horizon of arid sites (11‰) was very close to the parent material, while $\delta^{18}O_P$ values in the A horizon were higher (16–17‰), but still below the equilibrium range (Supplementary Table 6). The $\delta^{18}O_P$ of microbial P decreased to below equilibrium with increasing

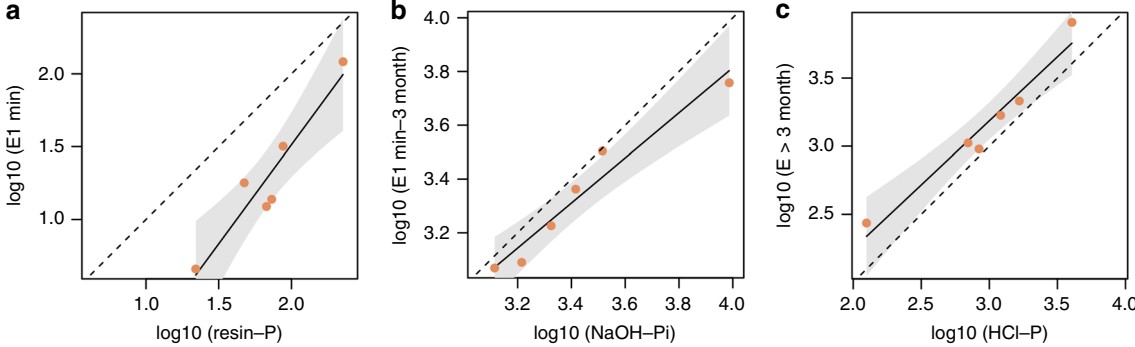

**Fig. 3** Correlation between chemically extracted P pools and isotopically exchangeable P. Dashed lines represent the 1:1 line; shaded areas the 95% confidence regions of the regression curves. Resin-P correlated with P exchangeable within 1 min (**a**), NaOH-Pi with P exchangeable within 3 months, and HCl-P with P that is only exchangeable at time spans longer than 3 months (**b**). $R^2$ values were 0.9, 0.95, and 0.95, respectively, with all $F$-statistics >34 and $p < 0.005$

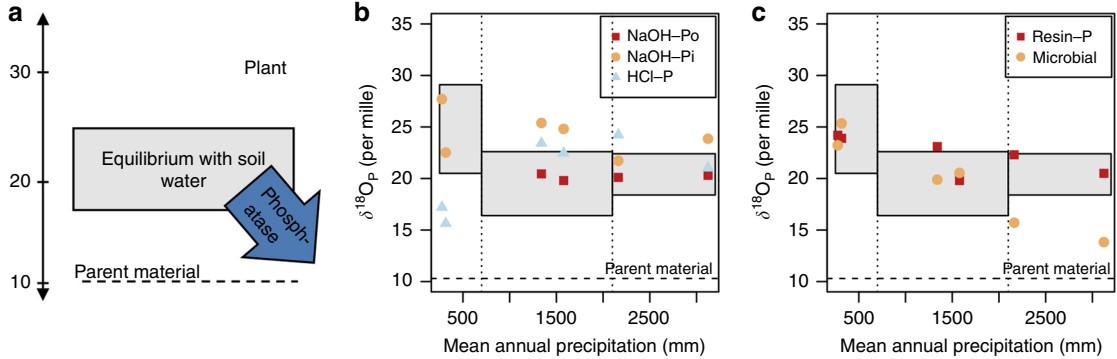

**Fig. 4** Oxygen-stable isotopes in soil phosphate pools. $\delta^{18}O_P$ of four processes: parent material and phosphatase activity lead to a below-equilibrium value, while plants are a source of heavier phosphates (**a**). $\delta^{18}O_P$ in soil pools: NaOH-Po, NaOH-Pi, and HCl-P (**b**) and resin-P and microbial P (**c**). The equilibrium range was calculated for each climatic zone (arid, subhumid, and humid). See Methods for details

precipitation, reaching 14–16‰ in humid sites. Microbial P $\delta^{18}O_P$ values increasingly diverged from resin-P $\delta^{18}O_P$ along the climatic gradient.

## Discussion

The power of the multi-method approach is exemplified in the analysis of the HCl-extractable P pool. In sequential extractions, the 1 M HCl-extractable pool is traditionally interpreted to comprise scarcely soluble Ca–P and assumed to be primary mineral P[6,32]. Our $\delta^{18}O_P$ data show that P in the HCl pool has been biologically recycled (at least partially in all but the B horizon of arid sites), the isotope exchange kinetics show that this P can arrive in solution, even after very long time periods, and the spectroscopic analysis did not detect apatite on wet sites. Hence, sequential extractions by themselves give only very static and limited information. Our approach and results allow making the following interpretations. In the subhumid and humid sites, during the 150,000 years of pedogenesis, all primary apatite has been dissolved; P released by weathering is modified by microbes and plants, and redistributed in different forms (organic, adsorbed, and re-precipitated) in the soil. These different forms remain in a dynamic equilibrium, since the exchanges between the pools continue and plants and microbes continue to incorporate P ions rich in $^{18}O$.

We used our data from exchange kinetic experiments and natural abundance of $^{18}O$ in phosphate to provide the first robust approximation of turnover times of commonly considered inorganic P pools. Turnover of dissolved P was recently analyzed in over 200 soils[12], and microbial P is known to turn over in a time frame of days to weeks[35,36], but information on turnover of other P pools was previously lacking. The correlation between chemically extracted pools and P pools defined by kinetics (Fig. 3), suggests that these pools describe the same fraction of soil P. Assuming that the exchange time necessary for a phosphate ion in solution to exchange with a phosphate ion located in a given pool present on the solid phase of the soil can be equated to the time necessary to renew the total amount of P present in this pool, then we can consider that this exchange time is similar to a turnover time. Following this logic, resin-P is completely exchanged within several minutes. Likewise, the correlation between NaOH-Pi and P exchangeable within 3 months and HCl-P and P exchangeable in >3 months suggests that the NaOH-Pi pool exchanges within several months, while the HCl-P only exchanges at longer time spans. The radioisotopic approach was complemented by $\delta^{18}O_P$, by making use of the incorporation of biogenic P into respective pools as a natural tracer for exchangeability at longer time spans. The fact that the NaOH-Pi pool (and the HCl-P pool on subhumid and humid sites) carried the biological signature, suggested that these pools are readily exchanged with the biosphere via the soil solution. Previously, it was shown that in a glacial forefield, HCl-P was primarily biogenic P after several thousand years of soil development[20]. Our study shows that under arid, high-pH conditions, geogenic P may persist in the HCl pool even after 150,000 years of soil development, i.e., that turnover of the HCl-P pool may be much slower

depending on climatic conditions. Based on these two approaches, we approximate that the resin-P, NaOH-Pi, and HCl-P pools turn over in minutes, weeks–months, and years–millennia, respectively.

While $\delta^{18}O_P$ of organic P is fundamental to calculate the release of phosphate by hydrolysis, our approach did not allow approximating the turnover of organic P pools. However, mean turnover of organic C on a Kohala site has been approximated to be 175,000 years[37]. Assuming that the turnover of organic P is linked to that of organic C, we can say that the majority of organic P is stabilized by amorphous minerals and effectively inaccessible to phosphatase enzymes, whereas a small fraction is available and characterized by rapid turnover.

In addition to providing information on turnover, measuring stable oxygen isotopes in soil and plant pools allowed differentiating between biologically and geochemically driven P fluxes. Several studies have used $\delta^{18}O_P$ to show that rapid cycling by pyrophosphatase, bringing $\delta^{18}O_P$ into equilibrium with soil water, dominates $\delta^{18}O_P$ of soil P pools[20,21,38]. In our study, $\delta^{18}O_P$ of NaOH-Pi and HCl-P in subhumid and humid sites tended to be higher than that of resin-P. Literature on fractionation effects by sorption/precipitation reactions is ambiguous but seems to suggest negligible effects at longer reaction times[39,40]. In our system, plant P was the only source identified with an above-soil equilibrium $\delta^{18}O_P$. Thus, we propose that in sites with high plant

productivity, some P from plants may be directly released into soil solution. From the soil solution, P may adsorb, precipitate, or stay in the resin-P pool. In the resin-P pool, P is more rapidly assimilated by microbes, which leads to the loss of the heavy $^{18}O$ signal, whereas the heavy $^{18}O$ signal may be retained longer in the NaOH-Pi and HCl-P pool. Alternatively, due to the longer turnover times of NaOH-Pi and HCl-P pools, these pools may also contain P that was cycled by microbes during times of lower temperature or water enriched in $^{18}O$, which would lead to a heavier $\delta^{18}O_P$[34].

At humid sites, $\delta^{18}O_P$ of microbial P was substantially lower than other soil pools, which could be due to isotopic differences between intracellular and extracellular water, intracellular hydrolysis of organic P, or a change in microbial community structure. A recent study showed that water within microbes may have a different isotopic composition than water outside of the cells[41], which implies that equilibrium $\delta^{18}O_P$ could be different from the one calculated using $\delta^{18}O$ of soil water. Alternatively, it has been shown that when P is in short supply, $\delta^{18}O_P$ of plant P decreased due to increased recycling of organic P by hydrolyzing enzymes within plant tissue[42]. We consider the most likely explanation for the lower $\delta^{18}O_P$ of microbial P at high rainfall sites to be tighter intramicrobial P recycling. As opposed to readily exchanging P with the environment as under more P-abundant conditions, at humid sites, P may be cycled more

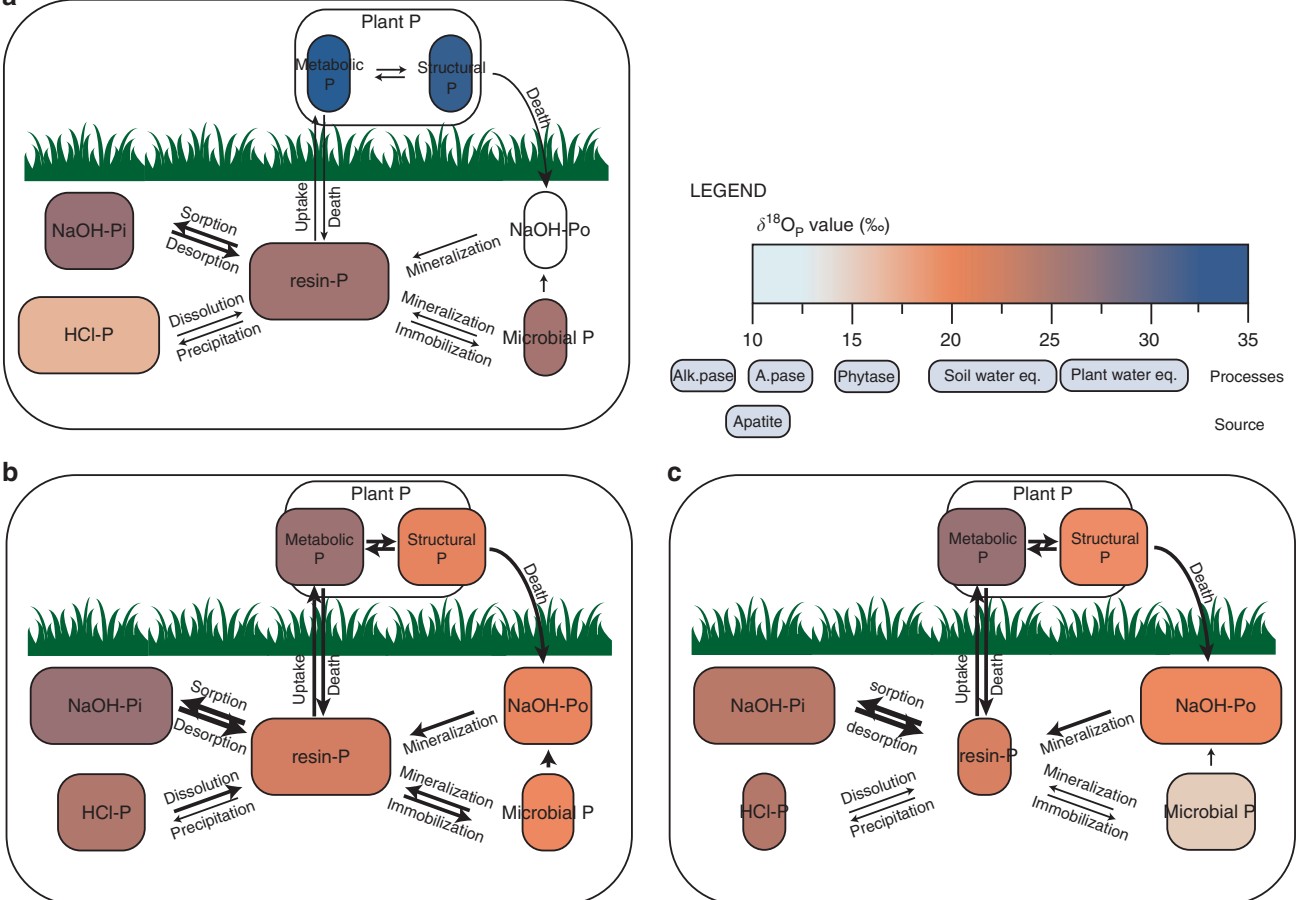

**Fig. 5** Phosphorus cycling in three different climatic zones. For each climatic zone, arid (**a**), subhumid (**b**), and humid (**c**), the size of boxes represents the relative size of the pool and the size of arrows represents the relative importance of the respective flux. Pool colors correspond to $\delta^{18}O_P$ values. We assumed four process signals: first, weathering of primary apatite with a $\delta^{18}O_P$ of 10‰; second, equilibration with water by pyrophosphatase in soil microbes (18–25‰); third, mineralization of organic P by hydrolyzing enzymes (7–17‰, depending on the enzyme), and fourth release of plant P (23–35‰) (Supplementary Tables 4 and 5). At arid sites, $\delta^{18}O_P$ of soil NaOH-Po could not be measured due to small concentrations

closely within the microbial community or microbes may increasingly take up organic P molecules and hydrolyze them within the cell.

Our approach also improves understanding of the effect of climate on P availability at the ecosystem scale. The Walker and Syer's model predicts declining P stocks with increasing weathering[43], but on the Kohala gradient, total P concentrations peaked at subhumid sites. A similar pattern was observed on a climatic gradient in northeastern China, where, as for the Kohala gradient, it was argued that biological uplift leads to higher P concentrations in surface soils at intermediate rainfall sites[26,44,45]. On the Kohala gradient, biological uplift is likely complemented by dust deposition, which increases with precipitation[28], and weathering rate, transferring P from rock to soil, which is the highest at subhumid sites[29]. P inputs are effectively retained due to high-surface-area minerals with a high P sorption capacity, inherent to these volcanic soils, allowing for the buildup of large amounts of exchangeable P. In humid sites, past weathering has depleted primary minerals and dust deposition and biological uplift are offset by leaching[26]. Total P and P availability is thus the highest on subhumid sites due to climate-dependent high P inputs and low outputs.

With increasing precipitation, P cycling was increasingly driven by biological as opposed to geochemical processes (Fig. 5). Indicators of inorganic P availability, such as water-extractable P and exchangeable P within 1 min, were considerably lower at high rainfall sites. In addition, the climate-driven changes in soil mineralogy led to increasing P sorption at high rainfall sites, as is shown by the low recovery of $^{33}P$ 1 min after addition. Increasing the competition between biota and sorption sites for available P may lead biota to increasingly rely on mineralization of organic P to meet their P requirements[46]. This is also expressed in the increase of potential acid phosphatase activity with increasing rainfall (Supplementary Fig. 3).

While P inputs through dust deposition and weathering are important for maintaining P fertility over time spans relevant for ecosystem development[13,14], we saw that they do not impact P availability at time spans relevant for plant growth. P deposition rate through dust is the highest at wettest sites, and approximated to be around $0.9\,mg\,P\,m^{-2}\,yr^{-1}$ (refs.[13,28]). Given the A-horizon magnitude of 10 cm and a bulk density of $0.40\,g\,cm^{-3}$ (ref.[27]), this translates to an input of $0.02\,mg\,P\,kg^{-1}\,yr^{-1}$ on a per-mass-A-horizon basis. When this value is juxtaposed to the per-minute flux of P between the solution and the solid phase ($14.2 \pm 0.74\,mg\,P\,kg^{-1}\,min^{-1}$, Supplementary Table 4), it becomes clear that dust cannot be expected to influence P availability in the short term, as internal exchange fluxes are orders of magnitude higher than external inputs, even on the wettest site. Similarly, the primary apatite $\delta^{18}O_P$ value was not found in any P pool relevant for plant nutrition, suggesting that processes other than weathering dominate short-term P availability. Rather, our data show that desorption and mineralization processes dominate short-term P availability.

In conclusion, combining spectroscopic and isotopic approaches improved our interpretation of sequential extractions and is key for process-based P modeling. Sequential extraction is probably the most commonly used method to study P distribution in soils, with thousands of studies using the Hedley method and its modifications[32,47]. By combining sequential extractions and spectroscopic analyses with isotopic approaches, we linked pools to exchange kinetics and were able to approximate turnover times of P pools, as well as trace P in different pools to biological and geochemical processes. Our process-based support of sequential extraction should thus aid in interpreting results from this common procedure. Furthermore, the integration of P cycling into Earth-system models has been slowed by an inadequate understanding of P dynamics in soils[1]. This also holds true in agricultural systems. P added as a fertilizer, once dissolved, is exposed to the same geochemical and biological processes as geogenic P and incorporated rapidly into different pools. Accordingly, fertilizer recovery over a growing season tends to be low, especially in highly P-sorbing soils, and most P taken up by plants stems from the soil reservoir or legacy fertilizer[48]. Tying the kinetic component and biological processes to the common representation of P pools for both natural and agroecosystems paves the way for model formulations that capture P dynamics in its many forms.

## Methods

**Site description and soil sampling**. The Kohala climatic gradient has been extensively described in previous works[26,27]. In brief, mean annual precipitation decreases on the leeward side of the Kohala volcano from around 3200 mm at around 1060 m above sea level to 250 mm at the coast[30]. Though the climate gradient also has a temperature component, precipitation has been viewed as the primary driver of soil processes and microbial community structure[26,27,49]. Priestley–Taylor potential evapotranspiration was derived from an interactive map of potential evapotranspiration of Hawai'i (Supplementary Table 1)[50]. Land use on all sites is grassland, though at low rainfall sites vegetation cover is sparse. For analysis of soil properties, modified Hedley extraction, isotope exchange kinetics, and oxygen-stable isotopes in phosphate, samples of A and B horizons were taken from six sites along a Hawaii lava flow on the Kohala volcano, Hawai'i in February 2016. Bulk P K-edge XANES was performed on four A-horizon samples sampled in May 2013; potential acid phosphatase activity was measured on 46 sites sampled in March 2015.

**Analysis of soil properties**. Soil pH was measured in water using a soil:solution ratio of 1:2.5 and 24-h equilibration time[51]. Total C and N were measured on bulk soil samples with an elemental analyzer (Vario Pyro Cube, Elementar GmbH, Hanau, Germany). Previously, subsamples were combusted (SSM-5000A, Shimadzu, Kyoto, Japan) and measured with an organic carbon analyzer (TOC-L, Shimadzu, Kyoto, Japan) to confirm that no inorganic C was present. The concentration of organic C was multiplied by 2 to get an estimate of organic matter concentration[52].

X-ray diffraction (XRD) was used to determine soil mineralogy, as well as the ratio of amorphous to crystalline fractions. First, the samples were finely ground using a McCrone Micronizing Mill (McCrone Scientific Ltd, London, UK) and then measured on a Bruker D8 Advance diffractometer using Cu Kα radiation and a high-resolution energy-dispersive 1D detector (Bruker AXS GmbH, Karlsruhe, Germany). To determine the amorphous matter content, samples were also measured after adding crystalline $Al_2O_3$ as internal standard. All diffractograms were analyzed with Rietveld quantitative-phase analysis using TOPAS (Bruker DIFFRAC.SUITE). Since amorphous fractions could contain both organic matter and amorphous inorganic materials, the concentration of organic matter was subtracted from the amorphous fraction to approximate the concentration of amorphous inorganic matter.

**Soil phosphorus pools**. A modified Hedley sequential extraction method was used to determine operationally defined inorganic and organic P pools[47]. The following extractions were performed in sequence on field-moist soil: anion exchange membrane (VWR International, Radnor, USA) with or without 0.54 M 1-hexanol, 0.25 M NaOH and 0.05 M EDTA, and 1 M HCl. The sorption-corrected difference between P extracted with hexanol and with only resin was taken as a proxy for microbial P[53]. In each fraction, inorganic P was measured by the malachite green method[54]. In the NaOH pool, organic fraction was determined by digestion with 2.5 M $H_2SO_4$ and 0.18 M $K_2S_2O_8$ at 110 °C for 60 min. This resulted in six operationally defined P pools: resin-P, microbial P, NaOH-Pi, NaOH-Porg, HCl-P, and residual P. Residual P was defined as total P, as measured by X-ray fluorescence (XEPOS, Spectro, Kleve, Germany), minus the sum of Hedley pools.

**Bulk P K-edge XANES**. To determine P classes, bulk P K-edge XANES was performed on A-horizon soil samples from sites 1, 4, 5, and 6 (Supplementary Table 1). Samples were homogenized, ground, and analyzed on Beamline 14-3 at the Stanford Synchrotron Radiation Lightsource (SSRL). Energy calibration was achieved by setting the top of the primary peak of a lazulite P K-edge XANES spectrum to 2153.5 eV. Phosphorus speciation was determined by linear combination fits of the unknown spectra using the Athena software program[55] and three known references of the XANES spectra (Supplementary Fig. 2 and Supplementary Table 3). The reference spectra were apatite (collected at SSRL for this study), P adsorbed on hematite, and an Al-humic P complex[56]. We considered apatite P to represent the broader class of Ca–P species, P adsorbed on hematite to represent Fe–P species, and Al-humic P complex to represent Al–P and organic P species. Hematite-bound P was chosen to represent Fe–P species because it is a common

Fe-bearing mineral in these soils[27]. Also, the Fe-related spectral features of the hematite-bound P were a good fit for those in the soil spectra. Both $PO_4$ adsorbed to Al oxides, and an Al-humic P complex reference spectra fit the bulk P $K$-edge XANES spectra well. We selected the Al-humic P reference because we considered it likely that organic phosphate groups were chiefly adsorbed directly to Al-oxide surfaces[57]. This adsorption geometry likely makes organic P in these soils effectively indistinguishable from inorganic P adsorbed to metal oxides, at least using P $K$-edge XANES. Analyses were repeated on duplicate spectra of three samples, with the results varying <2%. Bulk P $K$-edge XANES results are highly dependent on the choice of the reference spectra and normalization parameters used in the fit, and in this case can be considered semiquantitative with an uncertainty of ±10%.

**Isotope exchange kinetics.** Isotope exchange kinetics were determined using the method of Fardeau et al.[16]. In brief, 10 g of air-dried soil were added to 99 ml of water and shaken in an overhead shaker for 16 h. Then, 1 ml of carrier-free 0.09–0.26 MBq $^{33}P$ was added to the soil solution. Water-extractions were performed following the protocol of Frossard et al.[58]. The measured decline in radioactivity was modeled with

$$\frac{r_{(t)}}{R} = m\left(t + m^{\frac{1}{n}}\right)^{-n} + \frac{r_{(\infty)}}{R} \quad (4)$$

by fitting $m$ and $n$, where $r_{(t)}$ is the radioactivity measured at time $t$, and $R$ is the total amount of radioactivity added to the solution (Supplementary Figs. 4 and 5)[16,58]. The end-behavior of Eq. 4 was calculated as

$$\frac{r_{(\infty)}}{R} = \frac{P_w}{P_{inorg}} \quad (5)$$

where $P_w$ is the water-extractable P and $P_{inorg}$ is the total amount of inorganic P, which we set here equal to the sum of inorganic P pools from the Hedley extraction (resin + NaOH + HCl). Soil solution P turnover was calculated using

$$K_m = \frac{n}{m^{\frac{1}{n}}} \quad (6)$$

where $m$ and $n$ are parameters determined by fitting Eq. 4[12]. Isotopically exchangeable P ($E_{(t)}$), was calculated using Eq. 7

$$E_{(t)} = P_w \times \frac{R}{r_{(t)}} \quad (7)$$

where $R/r_{(t)}$ is taken from Eq. 4[33]. No microbial inhibitor was used, since in a pretest, addition of a microbial inhibitor did not affect $r_{(t)}/R$.

**Oxygen-stable isotopes in phosphate.** Oxygen-stable isotope ratios in phosphate were measured in resin, hexanol, inorganic NaOH, organic NaOH, and HCl-P pools of field-moist samples. For resin, hexanol, and HCl extracts, solutions were purified after sequential extraction[59]. For the HCl extraction, we controlled for possible hydrolysis by using $^{18}O$-labeled and unlabeled HCl solutions on splits of the samples. The NaOH pool was divided into high-molecular weight (HW) and low-molecular weight (LW) compounds using size exclusion gel chromatography prior to purification[60]. Extracts were passed through a gel column (ÄKTAprime plus, GE Healthcare Bio-Science AB, Uppsala, Sweden), where large molecules pass through the column quicker than smaller molecules[61]. Malachite-reactive and malachite-unreactive P was continuously measured on the effluent, where malachite-reactive P was assumed to be inorganic and malachite-unreactive P was assumed to be organic P (Supplementary Fig. 6). In this manner, the effluent could be divided into an organic (HW) and an inorganic (LW) pool. Any carryover of organic P into the inorganic fraction or vice versa was corrected for using mass-balance (Supplementary Table 7).

The LW pool was subjected to magnesium ammonium phosphate precipitation, which targets only inorganic P (Pi) and then purified following the standard protocol[59]. Two splits of the HW pool were hydrolyzed by UV radiation, with one split containing $^{18}O$-enriched $H_2O$ to check for possible hydrolysis[25]. Some P pools did not contain enough P to produce adequate $Ag_3PO_4$ for analysis.

$Ag_3PO_4$ samples were analyzed with three analytical replicates using a thermal conversion elemental analyzer (Vario Pyro Cube, Elementar GmbH), coupled to an isotopic ratio mass spectrometer (Isoprime 100, Elementar GmbH). All isotopic ratios are given in delta-notation relative to Vienna Standard Mean Ocean Water (VSMOW).

Equilibrium $\delta^{18}O_P$ was calculated for the time of sampling and for a potential equilibrium range using Eq. 3. Soil water was extracted using cryogenic vacuum extraction[62]. A more conservative, long-term equilibrium window was also calculated using soil water $\delta^{18}O$ and $T$ data from a previous study on the same climatic gradient[63]. First, a spread of $\delta^{18}O_{H2O}$ values was derived by calculating the mean plus or minus the standard deviation of 1 year of measurements. The spread was −0.95–5.93‰ for dry sites, −6.4–0.91‰ for subhumid, and −4.18–1.29‰ for wet sites. Second, ±$X$ was added to mean annual $T$ to get an approximation of $T$ fluctuation. For dry soils, $X$ was set to 5 °C, and for the remaining soils, $X = 2$ °C was used since these soils were deeper and $T$ fluctuation was thus less pronounced (see sampling depths in Supplementary Table 1). Third, Eq. 3 was applied, yielding

a $\delta^{18}O_P$ for dry sites of 20.5–29.1, for subhumid sites 16.4–22.6, and for humid sites 18.4–22.4‰.

The expected $\delta^{18}O_P$ from phosphate released by acid phosphatase, alkaline phosphatase, and phytase was calculated using enzyme-specific fractionation factors from the literature and the $\delta^{18}O_P$ of NaOH-Po in the soil[23–25]. It has been reported that apparent enzymatic isotope fractionations are substrate dependent, e.g., the hydrolysis of phytic acid can lead to a positive isotope fractionation[24]. However, for most enzyme–substrate combinations, isotope fractionations have been found to be negative[23–25], which suggests that the overall isotope effect caused by the enzymatic hydrolysis of organic P in the complex soil microbial environment is negative.

**Potential acid phosphatase activity.** Soils were sampled from 46 sites along the rainfall gradient and kept at 4 °C. Within 48 h, potential acid phosphatase activity was measured following the method of Saiya Cork et al.[64] using eight analytical replicates per sample. For regression fitting between potential enzymatic activity and mean annual precipitation, enzyme activity was log-transformed to meet the assumptions of normality. Because some enzyme activities were 0 and thus could not be log-transformed, the average standard deviation of enzyme activity (100 nmol $h^{-1}$ $g^{-1}$) was added to all observations prior to log-transformation.

**Plant analysis.** Dominant grass species were sampled on the six sites described in Table 1. Dominant grass species were buffel grass (*Pennisetum ciliare*) for the driest two sites and kikuyu grass (*Pennisetum clandestinum*) for the remaining four sites. At subhumid sites, perennial soybean (*Neonotonia wightii*) and at site 6 ohia trees (*Metrosideros polymorpha*) were also present.

Stable oxygen isotope ratios in phosphate were measured in TCA (trichloroacetic acid) and NaOH-extractable P, which we called metabolic and structural P, respectively[22]. The extractions were performed sequentially and in splits with $^{18}O$-labeled and unlabeled TCA. After the extraction with 0.3 M TCA, residual material was extracted using 0.25 M NaOH and 0.05 M EDTA[65] and hydrolyzed by UV radiation.

Plant water was extracted using cryogenic vacuum extraction[62]. Subsequently, Eq. 3 was used to calculate $\delta^{18}O_P$ in plants. T was set equal to air temperature at the time of sampling.

**Rock analysis.** Parent material was chipped from a road side-cut, where unweathered lava was exposed. To determine mineralogy, we followed the procedure using XRD outlined above for soils. To determine $\delta^{18}O_P$ of parent material, finely ground rock was dissolved with 1 M HCl, labeled and unlabeled, and purified according to protocol[59]. Parent material $\delta^{18}O_P$ for three analytical replicates was 10.3 ± 0.4‰.

**Data analysis.** All data analysis and plotting was performed in R[66].

**Data availability.** All data supporting the results reported in the article are included in this published article (and its supplementary information files).

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

## Acknowledgements

The project was funded by the Swiss National Science Foundation (Project number 200021_162422). Use of the Stanford Synchrotron Radiation Lightsource, SLAC National Accelerator Laboratory, is supported by the U.S. Department of Energy, Office of Science, and Office of Basic Energy Sciences under Contract No. DE-AC02-76SF00515. We acknowledge Dr. Laurie Schönholzer, Monika Macsai, Dr. Eva Meszaros, Charlotte Vermeiren, and Numa Pfenninger for lab help. S. Webb, C. Roach, and D. Day provided assistance with data collection at SSRL, and J. Allen aided in sample preparation for synchrotron analysis.

## Author contributions

Experiments were planned by J.H., F.T., E.F., O.A.C., and P.M.V. Measurement and analysis were carried out by J.H., F.T., C.V.S., M.S.M., C.P., and R.K. The manuscript was written by J.H. with input from all authors.

## Additional information

**Competing interests:** The authors declare no competing interests.

