## [Peer Review File · Nature Communications]

Reviewers' comments:

Reviewer #1 (Remarks to the Author):

Authors are commended on merging series of state of the art methods on phosphorus (P) speciation, turnover, and cycling to identify P cycling in 'a soil' (not in 'soil' because it is not generic) or even better 'a soil chronosequence'. Based on the various measurements, calculations, and assumptions, authors come to a conclusion that turnover time of P in soil is rather fast. While there is a significant challenge to compare results obtained from these methods, sufficient consideration of these limitations is needed prior to develop a generalized interpretation (at the level of confidence as expressed). Below I have listed experimental and methodological limitations, which limit the interpretation made from data and conclusions derived thereof.

Comments related to experimental and method limitations:

While the radioisotope dilution method has been used for a long, anything and everything done on this method so far is empirical. For example, there is one-minute extraction protocol- which can never be correct. First it is not possible to guarantee that the ^{33}P can be homogeneously dispersed in 1 minute and second solid and solution in a soil cannot be separated in 1 min (taking sample out and filtration or centrifugation needs more than 1 min). Someone 'should' correct this method. While I don't blame authors for using this method but I wondered why this issue has not alerted users enough and resonate expression that there is a sufficient limitation of using this method. Outcome of this experimental limitation is reflected in Fig 3, where there is 5 times or more offset between isotope vs chemically extracted results for exchangeable P pool, and there is no acceptable correlation between adsorbed P and exchangeable P pools. This warrants sufficient limitation of comparing results from these two methods.

Authors used known isotope fractionation factors for alkaline and acid phosphatase and phytase, but disregarded the fact that the fractionation factors cannot be generalized to an enzyme class because the enzyme from different sources and substrate from different sources have been found to be different. I understand this manuscript is not focused on that aspect nor there exist literature on all enzymes sources and substrates but appropriateness and uncertainty of fractionation factors and mentioning the risk of gross limitation of generalized has to be include in sufficient detail so that readers are aware of 'what if' questions.

Authors defined TCA extracted P as inorganic and NaOH extracted P as organic. But the fact is that the latter contains both organic and inorganic P in significant proportion in soils. Verification of inorganic P extracted by NaOH and accounting that P pool before hydrolyzing the organic P is needed prior to interpreting results generated from this method.

Authors used size exclusion chromatography to separated HW and LW organic fractions and assigned the HW as organic and LW as inorganic. While malachite method of testing LW as inorganic P is mentioned, I am not convinced that this is entirely true. The root of my suspicion comes from the fact that majority of organic P are small molecules and unless they are attached to larger organic compounds (in that case inorganic P should do the same), this classification has to have error of unknown magnitude. So rigorous data to support this claim has to be presented. This will also help future users to adopt this method with specific degree of confidence, if at all possible.

Phosphate oxygen isotope method of differentiating primary and secondary minerals is an appropriate methods but keep in mind that the chemical weathering of apatite, which is anticipated more in Hawaiian type chronosequence wont impact any isotope values unless they go for biological isotope exchange before precipitating again as secondary mineral.

Results and discussion:

With the experimental limitation mentioned above, I am skeptical whether the 'turnover' time extracted and expressed in the abstract and other parts of the manuscript is reliable. Well, this is not appropriate to point authors on the fallacies of a method that many authors have used (some

of the co-authors of this paper are forefront on the development of this method), but the authors should be cautious enough on the limitations of a method used and that caution has to be included into the equation weighing for interpreting results. This limitation is very well reflected in Fig 3. Therefore an in-depth analyses in the offset in Fig. 3 could provide additional reasons on the limitations mentioned above and alternatively could help authors to make an entirely different interpretation- which surely help both the comparison among methods and possibly identifying underlying processed based mechanism. Stable isotopes could be an alternative method to validate these results- which appears to be an expertise of this research team as well.

Reviewer #2 (Remarks to the Author):

This manuscript describes a new and innovative approach to investigating and quantifying the biogeochemical dynamics of phosphorus in soil-plant systems. It achieved this by selecting soils from a well characterized climosequence, and subjecting these soils to a unique combination of chemical analyses. To my knowledge this is the first time such an approach has been used to try and unravel the complexities of the bio/physic/chemical-properties and processes that drive the dynamics, bioavailability and mobility of phosphorus in terrestrial systems. This was possible due to the appropriate combination of contributors. The collective findings of the study confirm the importance of rainfall and associated weathering processes in driving the nature and dynamics of soil phosphorus, and while this may have been known previously, this is the first time to my knowledge that it has been quantified in a meaningful way. Given that these findings has global relevance and significance, I recommend that the manuscript be accepted for publication in Nature Communications. However, I do have one major comment for consideration by the authors. I accept that the combination of techniques used in this study facilitated improved understanding of phosphorus dynamics in an essentially undisturbed natural ecosystem, I would be grateful if the authors in their conclusions could consider if and how this approach could be used to similarly advance our understating of phosphorus dynamics in managed soil-plant systems (i.e. agroecosystems) where inputs, transfers and losses of phosphorus occur at elevated quantities compared with native ecosystems?

Reviewer #3 (Remarks to the Author):

This manuscript examines P chemistry in a climosequence of soils from Hawaii and applies of number of techniques to infer P cycling and the accumulation or depletion of specific P pools.

My major concern with this manuscript is the lack of novelty leading to a greater understanding of P turnover in soils. Despite the application of a range of established (and some outdated) methods to examine P forms in soil, the manuscript essentially comes to the same conclusions as other prior work (Chadwick, Feng, Walker and Syers) with little or no new insights. Using multiple methods to characterise element forms and behaviour in soil does not qualify the manuscript for Nature Communications in terms of novelty and originality¹.

The inclusion of sequential fractionation detracts from the manuscript quality - this technique from the 1950's ² has been overutilised in soil P research and there is sufficient evidence to question the interpretation of the data emanating from such analytical techniques³. The fact that sequential fractionation is commonly used is no recommendation that it provides insight into P forms and/or behaviour in soils.

Isotopic methods are also well established, as are kinetic methods to partition P into various pools (the first being McAuliffe in 19484), and some of the authors and others have already published on these in relation to P cycling in soils, in combination with XANES and/or examination of soil fractions.^{5, 6}

The stable oxygen isotope data is perhaps the most novel method employed, but suffers from multiple interpretations being possible to explain the isotopic shifts observed, so that in the end other methods are often used to help interpretation of ^{18}O data, rather than vice versa.

In places the manuscript reads more like a review rather than describing new insights from the analysis of this climosequence e.g. the section "Turnover" uses none of the data from this manuscript and is a review paragraph summarising results from multiple other studies. Indeed I believe this manuscript might be better rewritten as a review paper rather than an original contribution and submitted to a leading soil science journal.

1. Scheinost AC, Kretzschmar R, Pfister S, Roberts DR. Combining selective sequential extractions, X-ray absorption spectroscopy, and principal component analysis for quantitative zinc speciation in soil. *Environ Sci Technol* 36, 5021-5028 (2002).
2. Chang SC, Jackson ML. Soil phosphorus fractions in some representative soils. *Journal of Soil Science* 9, 109-119 (1958).
3. Negassa W, Leinweber P. How does the Hedley sequential phosphorus fractionation reflect impacts of land use and management on soil phosphorus: a review. *Journal of Plant Nutrition and Soil Science-Zeitschrift Fur Pflanzenernahrung Und Bodenkunde* 172, 305-325 (2009).
4. McAuliffe CD, Hall NS, Dean LA, Hendricks SB. Exchange reactions between phosphates and soils: Hydroxylic surfaces of soil minerals. *Soil Sci Soc Amer Proc* 12, 119-123 (1947).
5. Bunemann EK, et al. Rapid microbial phosphorus immobilization dominates gross phosphorus fluxes in a grassland soil with low inorganic phosphorus availability. *Soil Biol Biochem* 51, 84-95 (2012).
6. Beauchemin S, Hesterberg D, Chou J, Beauchemin M, Simard RR, Sayers DE. Speciation of phosphorus in phosphorus-enriched agricultural soils using X-ray absorption near-edge structure spectroscopy and chemical fractionation. *J Environ Qual* 32, 1809-1819 (2003).

Reviewer #4 (Remarks to the Author):

I think the paper is extremely relevant and timely, and that it directly approaches a difficult area of science by combining isotopic studies and XAS. I believe it is of potential interest to the readers of your journal, and indeed is potentially of great interest to all earth scientists.

In general, I felt that the findings of the authors were well supported; however the specifics of the XAS LCF analysis were somewhat confusing to me. The authors opted for a 3 component system that included an Al-bearing DOM as an organic standard. This was problematic as it was not totally clear whether PO_4 was sorbed to the organic ligands (which was implied as this is the only organic standard) or else as PO_4 adsorbed to short range order $\text{As}(\text{OH})_3$. To add to the confusion, the PO_4 on hematite standard chosen appears to have none of the pre-edge sp^3 to d orbital mixing characteristic of Fe oxide-phosphate complexation, and instead shows some character of phosphate salt XANES. I would assume this is due to the preparation method and counteracting present in that reference compound, but it is impossible to conclude that from the paper alone.

My experience suggests that the authors' choice of references may have somewhat biased their speciation conclusions, but that in the bigger picture this shouldn't preclude publication if they are willing to revise their models or their explanation to account for the issues that I explained above.

Response to reviewers

All line numbers refer to the line numbering in the clean manuscript.

Reviewer 1

Authors are commended on merging series of state of the art methods on phosphorus (P) speciation, turnover, and cycling to identify P cycling in 'a soil' (not in 'soil' because it is not generic) or even better 'a soil chronosequence'. Based on the various measurements, calculations, and assumptions, authors come to a conclusion that turnover time of P in soil is rather fast. While there is a significant challenge to compare results obtained from these methods, sufficient consideration of these limitations is needed prior to develop a generalized interpretation (at the level of confidence as expressed). Below I have listed experimental and methodological limitations, which limit the interpretation made from data and conclusions derived thereof.

Thank you for reviewing our manuscript and for the helpful comments. We addressed the methodological limitations as best as we could, as shown in the detailed response below.

We agree that the six soils on the Kohala climosequence are not generic soils. However, the Hawaiian system and this climatic gradient in particular has often been used as a model system (Chadwick et al. 2003, Vitousek 2004, Peay et al. 2017, von Sperber et al. 2017). As pointed out by reviewers 2 and 4, lessons learned from this study are useful for understanding soil P cycling in general. For this reason, we would like to keep the broad title, unless the editor is of a different opinion.

Comments related to experimental and method limitations:

While the radioisotope dilution method has been used for a long, anything and everything done on this method so far is empirical. For example, there is one-minute extraction protocol- which can never be correct. First it is not possible to guarantee that the ³³P can be homogeneously dispersed in 1 minute and second solid and solution in a soil cannot be separated in 1 min (taking sample out and filtration or centrifugation needs more than 1 min). Someone 'should' correct this method. While I don't blame authors for using this method but I wondered why this issue has not alerted users enough and resonate expression that there is a sufficient limitation of using this method.

As reviewer number 3 pointed out, isotopic methods using P radioisotopes are well-established with roots in the 1940s (McAuliffe et al. 1948). The isotope exchange kinetic (IEK) method was then further developed and tested by J.C. Fardeau and colleagues. Fardeau et al. used forward- and backward dilutions (Fardeau and Marini 1968) and tested multiple experimental variations (Fardeau and Jappe 1988) to show that the observed isotope dilution actually captures the mechanisms of P exchange between the soil particles and the solution. Unfortunately, these and further key publications validating the IEK method are in French, which explains why they are little known outside of French-speaking regions.

P exchangeable within 1 minute (E 1 min) has been shown to correspond to P in the soil solution and loosely-associated with soil particles using reverse radioisotope dilution: "After

determining the kinetic of isotopic exchange, Fardeau and Marini (1968) diluted the labeled suspension with a similar nonradioactive soil solution and measured immediately the new isotopic composition of P ions in solution. Quantifying this reverse isotopic dilution of the P ions in soil suspension, these authors show that a fraction of the P ions bound to the soil solid phase have the same rate of exchange as P ions in solution. The sum of this loosely-bound pool plus the quantity of P ions in solution gives the pool of the most mobile P ions (P_m) of the soil suspension.” (Morel et al. 2000, p. 54). These authors further showed, that this pool of most mobile P ions (P_m) was equal to E1min-values for a range of different soils.

IEK experiments are thus based on, and have done much to further, our understanding of exchange mechanisms. This explains why E-values derived from IEK perform much better at predicting crop response than other soil P tests (Frossard et al. 1994), and why E-values are widely accepted as the gold standard for determining P bioavailability (Hamon et al. 2002, Kruse et al. 2015).

To do a water-extraction in 1 minute, we followed the protocol described in Frossard et al. (2011): a small volume of soil-water suspension is taken up with a syringe and squeezed through a 0.2 µm filter. This takes only several seconds; the exact time in seconds of when the water-extract is filtered was noted and always within a few seconds of the anticipated measuring time. The fact that this works and produces robust and repeatable results is underlined by the raw data, which we have now added to the manuscript.

To address the reviewer’s concern, we have added the raw data of the radioisotope dilution experiments to the extended data (Extended Data Fig. 4 and 5). We also added information on the extraction protocol in the methods section (lines 412-413).

Outcome of this experimental limitation is reflected in Fig 3, where there is 5 times or more offset between isotope vs chemically extracted results for exchangeable P pool, and there is no acceptable correlation between adsorbed P and exchangeable P pools. This warrants sufficient limitation of comparing results from these two methods.

Our research into the offset in Fig 3 led us to discover a mistake in the data. The main offset is due to site 1, where we previously reported only 200 mg NaOH-extractable inorganic P. We redid the sequential extraction with three replicates and measured 1640 mg P/kg NaOH-extractable inorganic P. This value is more consistent with the other sites, as well as with measurements done by two independent master thesis works (unpublished). We regret that we did not spot this mistake in the previous version. This change affects Fig. 1a and Fig. 3b and c. The fits in Fig. 3b and 3c are much improved. However, the interpretations of the data are not affected by the correction of this error.

We have added the F-statistics, the p-values, and the R² of the simple linear regressions between isotopically exchangeable P and the chemically-extracted pools to the legend of Fig. 3.

The offset in Fig. 3a means that the E-1min value is an underestimation of resin extractable P. For example, if t is increased to 7, the points approach the 1:1 line (see Fig A below). Our conclusion that labile P turns over on the time scale of several minutes is thus not affected by

this offset. We prefer to keep E1min for the plot, because this value is often reported in the literature and because the E1min has been shown to comprise a homogeneous compartment of P in the soil solution and P loosely associated to soil particles (Morel et al. 2000).

Figure A. Correlations between labile P as determined by sequential extraction and E-values at $t = 1$ min (a) and $t = 7$ min (b). Units for all axes are $\log(\text{mg P}/\text{kg soil})$.

Authors used known isotope fractionation factors for alkaline and acid phosphatase and phytase, but disregarded the fact that the fractionation factors cannot be generalized to an enzyme class because the enzyme from different sources and substrate from different sources have been found to be different. I understand this manuscript is not focused on that aspect nor there exist literature on all enzymes sources and substrates but appropriateness and uncertainty of fractionation factors and mentioning the risk of gross limitation of generalized has to be include in sufficient detail so that readers are aware of ‘what if’ questions.

We thank the reviewer for this comment. We agree that it is not possible to test the fractionation factors of all enzymes and substrates present in the soil environment – this would be an impossible task. However, all phosphatases are enzymes that catalyze the transfer of a phosphoryl group from a phosphomonoester to water, which leads to the formation of inorganic phosphate and a negatively charged leaving group. This biochemical function is independent of the organism which synthesizes a phosphatase, because on an evolutionary time-scale it is most likely much older than most existing organisms. During the enzymatic hydrolysis of an organic P substrate, oxygen from water is incorporated into the newly formed inorganic phosphate and it is this incorporation which causes the observed isotope fractionation. For example, in the case of alkaline phosphatases, the incorporated oxygen is derived from a hydroxide ion (Kim and Wyckoff 1991, Stec et al. 2000) whereas in the case of acid phosphatase, the oxygen is directly derived from a water molecule (Lindqvist et al. 1994, Ortlund et al. 2003). The isotope fractionation depends first of all on the underlying reaction mechanisms rather than the organism (source) which produces the phosphatase. The observed isotope fractionation of all phosphatases that have been investigated under controlled laboratory conditions with most model substrates have been reported to be negative: approx. -10‰ for acid phosphatases and approx. -30‰ for alkaline phosphatases (Liang and Blake 2006, von Sperber et al. 2014).

We do agree with the reviewer that these isotope fractionations are not only dependent on the reaction mechanism but in some cases also on the organic P substrate. For example, it has been shown that the hydrolysis of phytic acid leads to a positive isotope fractionation (von Sperber et al. 2015, Wu et al. 2015). In the complex soil microbial environment, a multitude of enzyme-substrate combinations most likely occurs at the same time and it is impossible to disentangle every single one of these processes. As most enzyme-substrate combinations have been reported to cause negative isotope fractionations, we believe that the overall enzymatic isotope effect in the soil microbial environment is negative. We have added two sentences in the methods section to clarify our reasoning (lines 475-480).

Authors defined TCA extracted P as inorganic and NaOH extracted P as organic. But the fact is that the latter contains both organic and inorganic P in significant proportion in soils. Verification of inorganic P extracted by NaOH and accounting that P pool before hydrolyzing the organic P is needed prior to interpreting results generated from this method.

It seems that this comment arises from a misunderstanding. As is written in the section "plant analyses", a sequential extraction with TCA followed by NaOH was applied only on plant material, not on soils. The resulting pools were called metabolic and structural P, respectively, following recent literature (Pfahler et al. 2013, Noack et al. 2014, Pfahler et al. 2017). For separating inorganic and organic P in the soil NaOH pool, please see response below.

To prevent other readers from having this misunderstanding, we changed the subsection header in the methods section from "Phosphorus pools" to "Soil phosphorus pools".

Authors used size exclusion chromatography to separate HW and LW organic fractions and assigned the HW as organic and LW as inorganic. While malachite method of testing LW as inorganic P is mentioned, I am not convinced that this is entirely true. The root of my suspicion comes from the fact that majority of organic P are small molecules and unless they are attached to larger organic compounds (in that case inorganic P should do the same), this classification has to have error of unknown magnitude. So rigorous data to support this claim has to be presented. This will also help future users to adopt this method with specific degree of confidence, if at all possible.

Indeed, this study would be the first to report stable oxygen isotopes values in organic phosphate, which was previously not possible due to methodological limitations. We have submitted a methods paper to European Journal of Soil Science delineating this method carefully. This method paper was recently accepted (10.4.2018), and should be available online soon.*

* Tamburini, F, Pistocchi, C, Helfenstein, J, Frossard, E. A method to analyse the isotopic composition of oxygen associated to organic phosphorus in soil and plant material. *European Journal of Soil Science* (accepted).

To improve the description of the SEGC step in the manuscript, we added plots of the elution curves (Extended Data Fig. 6) and a table (Extended Data Table 7) with the amount of total and malachite-reactive P in each pool after size-separation. We also amended the methods section to outline the methods used more clearly, also citing Jarosch et al. 2015, a previous study using SEGC to separate the NaOH-pool into inorganic and organic fractions (Jarosch et al. 2015) (lines 436-445). Finally, we propose to cite the EJSS methods paper as soon as it is published.

Phosphate oxygen isotope method of differentiating primary and secondary minerals is an appropriate methods but keep in mind that the chemical weathering of apatite, which is anticipated more in Hawaiian type chronosequence wont impact any isotope values unless they go for biological isotope exchange before precipitating again as secondary mineral.

We fully agree with this comment and upon re-reading the manuscript we do see that this may not have been clear to the reader. In the introduction it is clearly stated, “under soil conditions, only enzymes can break the bond between P and O in phosphate” (line 77) and we made an effort to remove any confusing sentences in the revised discussion.

Results and discussion:

With the experimental limitation mentioned above, I am skeptical whether the ‘turnover’ time extracted and expressed in the abstract and other parts of the manuscript is reliable. Well, this is not appropriate to point authors on the fallacies of a method that many authors have used (some of the co-authors of this paper are forefront on the development of this method), but the authors should be cautious enough on the limitations of a method used and that caution has to be included into the equation weighing for interpreting results. This limitation is very well reflected in Fig 3. Therefore an in-depth analyses in the offset in Fig. 3 could provide additional reasons on the limitations mentioned above and alternatively could help authors to make an entirely different interpretation- which surely help both the comparison among methods and possibly identifying underlying processed based mechanism. Stable isotopes could be an alternative method to validate these results- which appears to be an expertise of this research team as well.

We have addressed the experimental limitations (see responses above), added missing supplementary information to undermine our analyses, and expounded on methods used. We have also addressed the offset in Fig 3. These modifications strengthen our interpretation of turnover times for the different pools. While this is the first study providing estimates of turnover times of all these pools, our estimates are in line with previous evidence, e.g. from studies tracing the incorporation of ³³P or ¹⁸O into different P pools (Buehler et al. 2002, Bünemann et al. 2004, Vu et al. 2009, Tamburini et al. 2012, Joshi et al. 2016). Hence, we consider our order-of-magnitude estimates for turnover times as robust and reliable.

Reviewer 2

This manuscript describes a new and innovative approach to investigating and quantifying the biogeochemical dynamics of phosphorus in soil-plant systems. It achieved this by selecting soils from a well characterized climosequence, and subjecting these soils to a

unique combination of chemical analyses. To my knowledge this is the first time such an approach has been used to try and unravel the complexities of the bio/physic/chemical-properties and processes that drive the dynamics, bioavailability and mobility of phosphorus in terrestrial systems. This was possible due to the appropriate combination of contributors. The collective findings of the study confirm the importance of rainfall and associated weathering processes in driving the nature and dynamics of soil phosphorus, and while this may have been known previously, this is the first time to my knowledge that it has been quantified in a meaningful way. Given that these findings has global relevance and significance, I recommend that the manuscript be accepted for publication in Nature Communications. However, I do have one major comment for consideration by the authors. I accept that the combination of techniques used in this study facilitated improved understanding of phosphorus dynamics in an essentially undisturbed natural ecosystem, I would be grateful if the authors in their conclusions could consider if and how this approach could be used to similarly advance our understating of phosphorus dynamics in managed soil-plant systems (i.e. agroecosystems) where inputs, transfers and losses of phosphorus occur at elevated quantities compared with native ecosystems?

We thank the reviewer for their positive comments on our manuscript. We have added several sentences discussing relevance for agroecosystems in the “implications” paragraph (lines 329-336).

Reviewer 3

This manuscript examines P chemistry in a climosequence of soils from Hawaii and applies of number of techniques to infer P cycling and the accumulation or depletion of specific P pools.

My major concern with this manuscript is the lack of novelty leading to a greater understanding of P turnover in soils. Despite the application of a range of established (and some outdated) methods to examine P forms in soil, the manuscript essentially comes to the same conclusions as other prior work (Chadwick, Feng, Walker and Syers) with little or no new insights. Using multiple methods to characterise element forms and behaviour in soil does not qualify the manuscript for Nature Communications in terms of novelty and originality¹.

We thank the reviewer for their constructive feedback. While a lot of previous studies have claimed to provide insights on P dynamics, this was actually not possible given the methods that were applied. The studies mentioned by the reviewer (Chadwick, Feng, Walker and Syers) relied solely on sequential extractions. However, sequential extraction has many limitations and can only provide information on P stocks. As was written in the recent Marschner review in Plant and Soil, “an improved conceptual model of P cycling in soils is needed...in this new conceptual model of P cycling, a temporal (kinetic) component of soil P transformations must be considered” (Menezes-Blackburn et al. 2017). Capturing actual dynamics requires using isotopes, which can trace fluxes between different pools. This article is the first where both radioisotopes and stable isotopes have been used in the soil-system to assess P dynamics; the conclusions on turnover of different P pools are the most robust of

their kind. We would like to refer to reviewers 2 and 4, who both stated that the global relevance and novelty of this manuscript warrant publication in Nature Communications.

To address the reviewer's comments, we have tried to better point out the novelty of this work in the abstract and the first paragraph of the discussion.

The inclusion of sequential fractionation detracts from the manuscript quality - this technique from the 1950's has been overutilised in soil P research and there is sufficient evidence to question the interpretation of the data emanating from such analytical techniques³. The fact that sequential fractionation is commonly used is no recommendation that it provides insight into P forms and/or behaviour in soils.

We agree that sequential fractionation has limitations because it separates P into operationally rather than functionally defined pools. Unfortunately, there is still no alternative to sequential extraction to assess soil P stocks as a whole. While ³¹P NMR is useful for examining organic P species, and XANES can capture inorganic P species, sequential extraction remains the only single technique able to provide information on inorganic forms and organic forms as well as smaller pools such as labile and microbial P. Rather, authors of the recent article on innovative methods in P research suggest, that different methods be combined according to their individual strengths and limitations (Kruse et al. 2015). We addressed the limitations of the sequential extraction by cross-validating sequential extraction pools (where possible) with two completely independent methods: XANES and isotopic exchange kinetics. An interesting component of this manuscript is that despite its limitations, sequential fractionation results are relatively consistent with XANES (fig. 1) and highly correlated to isotope exchange kinetic pools (fig. 3). This finding not only supports our use of sequential extractions, but it is also of interest to help in the interpretation of sequential fractionation results for future work.

Isotopic methods are also well established, as are kinetic methods to partition P into various pools (the first being McAuliffe in 1948), and some of the authors and others have already published on these in relation to P cycling in soils, in combination with XANES and/or examination of soil fractions.^{5, 6}

We agree with the reviewer that radioisotopic methods have been used for a long time in P research. However, McAuliffe and the other studies cited by the reviewer only considered specific components of P cycling. McAuliffe et al. (1948) studied short-term (30 minutes) abiotic exchange processes. (Bünemann et al. 2012) specifically looked at microbial processes at short temporal scales, but did not consider geochemical processes or plants. Likewise, the (Beauchemin et al. 2003) paper also cited by the reviewer considered speciation, but did not address fluxes and processes. Our manuscript is the first holistic attempt to bring together pools, speciation, biological and geochemical fluxes and processes on one model ecosystem. This only becomes possible when considering spectroscopic and isotopic techniques.

To better underline the power of our approach, we added a new paragraph to showcase the example of HCl-P (lines 201-213).

The stable oxygen isotope data is perhaps the most novel method employed, but suffers from multiple interpretations being possible to explain the isotopic shifts observed, so that in the end other methods are often used to help interpretation of ^{18}O data, rather than vice versa.

Like with any new method, there is indeed more uncertainty in interpretation of ^{18}O data compared to more established methods. From our point of view it is a strength of this paper and not a weakness that other methods are used to underpin the results and reduce the uncertainty.

In places the manuscript reads more like a review rather than describing new insights from the analysis of this climosequence e.g. the section "Turnover" uses none of the data from this manuscript and is a review paragraph summarising results from multiple other studies. Indeed I believe this manuscript might be better rewritten as a review paper rather than an original contribution and submitted to a leading soil science journal.

We do not agree with the reviewer that the article should be rewritten as a review because we present a wide range of new data. We revised the "Turnover" section to better underline that it relies directly on two independent sources of evidence on P pool turnover: 1) the correlation between the sequential extraction pools, and 2) the stable oxygen isotopes in phosphate (lines 263-290).

1. Scheinost AC, Kretzschmar R, Pfister S, Roberts DR. Combining selective sequential extractions, X-ray absorption spectroscopy, and principal component analysis for quantitative zinc speciation in soil. *Environ Sci Technol* 36, 5021-5028 (2002).
2. Chang SC, Jackson ML. Soil phosphorus fractions in some representative soils. *Journal of Soil Science* 9, 109-119 (1958).
3. Negassa W, Leinweber P. How does the Hedley sequential phosphorus fractionation reflect impacts of land use and management on soil phosphorus: a review. *Journal of Plant Nutrition and Soil Science-Zeitschrift Fur Pflanzenernahrung Und Bodenkunde* 172, 305-325 (2009).
4. McAuliffe CD, Hall NS, Dean LA, Hendricks SB. Exchange reactions between phosphates and soils: Hydroxylic surfaces of soil minerals. *Soil Sci Soc Amer Proc* 12, 119-123 (1947).
5. Bunemann EK, et al. Rapid microbial phosphorus immobilization dominates gross phosphorus fluxes in a grassland soil with low inorganic phosphorus availability. *Soil Biol Biochem* 51, 84-95 (2012).
6. Beauchemin S, Hesterberg D, Chou J, Beauchemin M, Simard RR, Sayers DE. Speciation of phosphorus in phosphorus-enriched agricultural soils using X-ray absorption near-edge structure spectroscopy and chemical fractionation. *J Environ Qual* 32, 1809-1819 (2003).

Reviewer 4

I think the paper is extremely relevant and timely, and that it directly approaches a difficult area of science by combining isotopic studies and XAS. I believe it is of potential interest to the readers of your journal, and indeed is potentially of great interest to all earth scientists.

We thank the reviewer for the positive comments.

In general, I felt that the findings of the authors were well supported; however the specifics of the XAS LCF analysis were somewhat confusing to me. The authors opted for a 3 component system that included an Al-bearing DOM as an organic standard. This was problematic as it was not totally clear whether PO₄ was sorbed to the organic ligands (which was implied as this is the only organic standard) or else as PO₄ adsorbed to short range order As(OH)₃. To add to the confusion, the PO₄ on hematite standard chosen appears to have none of the pre-edge sp³ to d orbital mixing characteristic of Fe oxide-phosphate complexation, and instead shows some character of phosphate salt XANES. I would assume this is due to the preparation method and counterings present in that reference compound, but it is impossible to conclude that from the paper alone.

My experience suggests that the authors' choice of references may have somewhat biased their speciation conclusions, but that in the bigger picture this shouldn't preclude publication if they are willing to revise their models or their explanation to account for the issues that I explained above.

The Reviewer pointed out two separate issues, which we will address in order; we describe the revisions to the manuscript that these considerations entailed below:

- 1) ***“Organic” P fitting standard:*** *Unfortunately, using multiple types of spectroscopic analysis ranging from P K-edge micro-XANES spectroscopy, to infrared spectroscopy, to P L-edge spectroscopy, we have found few explicit clues regarding organic P speciation in these soils (and what evidence we have will likely be presented in detail in another contribution, as detailed discussion seems beyond the scope of the present work). There is some spectroscopic evidence that organic P may be partly in phosphonate form, but including a phosphonate spectrum in bulk P K-edge XANES fits did not yield useful results.*

In contrast, spectra from both PO₄ adsorbed to Al oxides, and an Al oxide/humic P complex give excellent fits to the bulk P K-edge XANES spectra presented here. From the available spectroscopic evidence, it seems likely that organic phosphate groups may have been chiefly adsorbed directly to Al oxide surfaces, occupying the “contact zone” in the “zonal” model of Kleber et al. (2007) (Kleber et al. 2007). This may make organic P in these soils effectively indistinguishable from inorganic P adsorbed to Al oxides, at least using P K-edge XANES.

Uncertainty arising from choice of “standard” spectra was included in the uncertainty

estimate, and indeed was the chief component of the uncertainty estimate.

- 2) ***PO₄ on hematite:*** The Reviewer noted the characteristic pre-edge feature that is diagnostic of the presence of Fe-associated P. Though it is subtle, the Fe-associated pre-edge feature is present in both the “standard” spectrum and the “unknown” spectra (where noted). For an example, see the included spectral plots. The “PO₄ adsorbed to hematite” spectrum was used in the fit because other standard spectra (e.g., for PO₄ adsorbed to goethite) did not yield particularly good fits. Hematite is an important component of the soil mineralogy at these sites (Chadwick et al. 2003), so PO₄ on hematite seems to be a reasonable model spectrum to represent Fe-associated P (which, the spectra indicate, is an important component).

Figure A. Phosphorus K-edge XANES spectra of Kohala soil (blue), compared to phosphate adsorbed on Al oxide from Giguët-Covex et al. (Giguët-Covex et al. 2013)(red). The pre-edge feature indicative of Fe-associated P is very subtle, but clearly evident around 2150 eV in the Kohala soil spectrum. Replicate measurements at two synchrotron facilities on separate continents show this pre-edge feature in the spectra, and fitting spectra were chosen to help quantify this feature.

Revisions in response to Reviewer comments

- 1) ***“Organic” P fitting standard:*** We added some explanation and clarification regarding P speciation, organic P adsorption, and the limitations of P K-edge XANES in distinguishing between P species to the methods section (lines 392-403).
- 2) ***PO₄ on hematite:*** We added to the methods section that hematite was identified as an important component of the soils using XRD, and that we thus think it is a

reasonable model spectrum to represent Fe-associated P. Since the feature the Reviewer noted was present in the soil spectra (see Figure A) and explicitly accounted for in the choice of standard spectra and fitting results, we hope no further action is required to allay the Reviewer's concerns in this case (lines 392-403).

References

- Beauchemin, S., D. Hesterberg, J. Chou, M. Beauchemin, R. Simard, and D. Sayers. 2003. Speciation of Phosphorus in Phosphorus-Enriched Agricultural Soils Using X-Ray Absorption Near-Edge Structure Spectroscopy and Chemical Fractionation. *Journal of Environment Quality* **32**:1809-1819.
- Buehler, S., A. Oberson, I. M. Rao, D. K. Friesen, and E. Frossard. 2002. Sequential Phosphorus Extraction of a ³³P-Labeled Oxisol under Contrasting Agricultural Systems. *Soil Science Society of America Journal* **66**:868-877.
- Bünemann, E. K., a. Oberson, F. Liebisch, F. Keller, K. E. Annaheim, O. Huguenin-Elie, and E. Frossard. 2012. Rapid microbial phosphorus immobilization dominates gross phosphorus fluxes in a grassland soil with low inorganic phosphorus availability. *Soil Biology and Biochemistry* **51**:84-95.
- Bünemann, E. K., F. Steinebrunner, P. C. Smithson, E. Frossard, and A. Oberson. 2004. Phosphorus Dynamics in a Highly Weathered Soil as Revealed by Isotopic Labeling Techniques. *Soil Science Society of America Journal* **68**:1645-1655.
- Chadwick, O. A., R. T. Gavenda, E. F. Kelly, K. Ziegler, C. G. Olson, W. Crawford Elliott, and D. M. Hendricks. 2003. The impact of climate on the biogeochemical functioning of volcanic soils. *Chemical Geology* **202**:195-223.
- Fardeau, J. C., and J. Jappe. 1988. Valeurs caractéristique des cinétiques de dilution isotopique des ions phosphate dans les systèmes sol-solution. Pages 79-99 *in* L. Gachon, editor. *Phosphore et potassium dans les relations sol-plante*. Institute National de la Recherche Agronomique, Paris.
- Fardeau, J. C., and P. Marini. 1968. Détermination, par échange isotopique en retour, du compartiment des ions-phosphates les plus mobiles du sol. *C.R. Acad. Sci. Paris*.
- Frossard, E., J. L. Morel, J. C. Fardeau, and M. Brossard. 1994. Soil isotopically exchangeable phosphorus: A comparison between E and L values. *Soil Science Society of America Journal* **58**:846-851.
- Giguet-Covex, C., J. Poulencard, E. Chalmin, F. Arnaud, C. Rivard, J. P. Jenny, and J. M. Dorioz. 2013. XANES spectroscopy as a tool to trace phosphorus transformation during soil genesis and mountain ecosystem development from lake sediments. *Geochimica et Cosmochimica Acta* **118**:129-147.
- Hamon, R. E., I. Bertrand, and M. J. McLaughlin. 2002. Use and abuse of isotopic exchange data in soil chemistry. *Soil Research* **40**:1371-1381.
- Jarosch, K. A., A. L. Doolette, R. J. Smernik, F. Tamburini, E. Frossard, and E. K. Bünemann. 2015. Characterisation of soil organic phosphorus in NaOH-EDTA extracts: A comparison of ³¹P NMR spectroscopy and enzyme addition assays. *Soil Biology and Biochemistry* **91**:298-309.
- Joshi, S. R., X. Li, and D. P. Jaisi. 2016. Transformation of Phosphorus Pools in an Agricultural Soil: An Application of Oxygen-18 Labeling in Phosphate. *Soil Science Society of America Journal* **80**:69.

- Kim, E. E., and H. W. Wyckoff. 1991. Reaction mechanism of alkaline phosphatase based on crystal structures: Two-metal ion catalysis. *Journal of Molecular Biology* **218**:449-464.
- Kleber, M., P. Sollins, and R. Sutton. 2007. A conceptual model of organo-mineral interactions in soils: self-assembly of organic molecular fragments into zonal structures on mineral surfaces. *Biogeochemistry* **85**:9-24.
- Kruse, J., M. Abraham, W. Amelung, C. Baum, R. Bol, O. Kühn, H. Lewandowski, J. Niederberger, Y. Oelmann, C. Rüger, J. Santner, M. Siebers, N. Siebers, M. Spohn, J. Vestergren, A. Vogts, and P. Leinweber. 2015. Innovative methods in soil phosphorus research: A review. *Journal of Plant Nutrition and Soil Science* **178**:43-88.
- Liang, Y., and R. E. Blake. 2006. Oxygen isotope signature of Pi regeneration from organic compounds by phosphomonoesterases and photooxidation. *Geochimica et Cosmochimica Acta* **70**:3957-3969.
- Lindqvist, Y., G. Schneider, and P. Vihko. 1994. Crystal structures of rat acid phosphatase complexed with the transition-state analogs vanadate and molybdate. *European Journal of Biochemistry* **221**:139-142.
- McAuliffe, C. D., N. S. Hall, L. A. Dean, and S. B. Hendricks. 1948. Exchange reactions between phosphate and soils: hydroxylic surfaces of soil minerals. *Soil Science Society of America Proceedings* **12**:119-123.
- Menezes-Blackburn, D., C. Giles, T. Darch, T. S. George, M. Blackwell, M. Stutter, C. Shand, D. Lumsdon, P. Cooper, R. Wendler, L. Brown, D. S. Almeida, C. Wearing, H. Zhang, and P. M. Haygarth. 2017. Opportunities for mobilizing recalcitrant phosphorus from agricultural soils: a review. *Plant and Soil*.
- Morel, C., H. Tunney, D. Plénet, and S. Pellerin. 2000. Transfer of phosphate ions between soil and solution: Perspectives in soil testing. *Journal of Environmental Quality* **29**:50-59.
- Noack, S. R., M. J. McLaughlin, R. J. Smernik, T. M. McBeath, and R. D. Armstrong. 2014. Phosphorus speciation in mature wheat and canola plants as affected by phosphorus supply. *Plant and Soil* **378**:125-137.
- Ortlund, E., M. W. LaCount, and L. Lebioda. 2003. Crystal Structures of Human Prostatic Acid Phosphatase in Complex with a Phosphate Ion and α -Benzylaminobenzylphosphonic Acid Update the Mechanistic Picture and Offer New Insights into Inhibitor Design. *Biochemistry* **42**:383-389.
- Peay, K. G., C. von Sperber, E. Cardarelli, H. Toju, C. A. Francis, O. A. Chadwick, and P. M. Vitousek. 2017. Convergence and contrast in the community structure of Bacteria, Fungi and Archaea along a tropical elevation–climate gradient. *FEMS Microbiology Ecology* **93**.
- Pfahler, V., T. Dürr-Auster, F. Tamburini, M. S. Bernasconi, and E. Frossard. 2013. ¹⁸O enrichment in phosphorus pools extracted from soybean leaves. *The New phytologist* **197**:186-193.
- Pfahler, V., F. Tamburini, S. M. Bernasconi, and E. Frossard. 2017. A dual isotopic approach using radioactive phosphorus and the isotopic composition of oxygen associated to phosphorus to understand plant reaction to a change in P nutrition. *Plant Methods* **13**:75.
- Stec, B., K. M. Holtz, and E. R. Kantrowitz. 2000. A revised mechanism for the alkaline phosphatase reaction involving three metal ions¹¹Edited by R. Huber. *Journal of Molecular Biology* **299**:1303-1311.

- Tamburini, F., V. Pfahler, E. K. Bünemann, K. Guelland, S. M. Bernasconi, and E. Frossard. 2012. Oxygen isotopes unravel the role of microorganisms in phosphate cycling in soils. *Environmental Science and Technology* **46**:5956-5962.
- Vitousek, P. M. 2004. *Nutrient Cycling and Limitation: Hawai'i as a Model System*. Princeton University Press, Princeton.
- von Sperber, C., O. A. Chadwick, K. L. Casciotti, K. G. Peay, C. A. Francis, A. E. Kim, and P. M. Vitousek. 2017. Controls of nitrogen cycling evaluated along a well-characterized climate gradient. *Ecology*.
- von Sperber, C., H. Kries, F. Tamburini, S. M. Bernasconi, and E. Frossard. 2014. The effect of phosphomonoesterases on the oxygen isotope composition of phosphate. *Geochimica et Cosmochimica Acta* **125**:519-527.
- von Sperber, C., F. Tamburini, B. Brunner, S. M. Bernasconi, and E. Frossard. 2015. The oxygen isotope composition of phosphate released from phytic acid by the activity of wheat and *Aspergillus niger* phytase. *Biogeosciences* **12**:4175-4184.
- Vu, D. T., C. Tang, and R. D. Armstrong. 2009. Transformations and availability of phosphorus in three contrasting soil types from native and farming systems: A study using fractionation and isotopic labeling techniques. *Journal of Soils and Sediments* **10**:18-29.
- Wu, J., P. Paudel, M. Sun, S. R. Joshi, L. M. Stout, R. Greiner, and D. P. Jaisi. 2015. Mechanisms and Pathways of Phytate Degradation: Evidence from Oxygen Isotope Ratios of Phosphate, HPLC, and Phosphorus-31 NMR Spectroscopy. *Soil Science Society of America Journal* **79**:1615-1628.

REVIEWERS' COMMENTS:

Reviewer #1 (Remarks to the Author):

Authors have responded my comments satisfactorily. So I recommend to publish this manuscript.

Reviewer #4 (Remarks to the Author):

In my opinion, the authors have addressed my technical concerns regarding this manuscript, and also seem to have carefully addressed the issues from other reviewers in the revisions. I therefore feel that the manuscript should be accepted. I think this manuscript will push the scientific understanding of soil P cycling and will be an important contribution to the soil science and biogeochemistry communities.